# Character configuration, major depressive episodes, and suicide-related ideation among Japanese undergraduates

Keisuke Takanobu[1], Nobuyuki Mitsui[1]*, Shinya Watanabe[1],
Kuniyoshi Toyoshima[1], Yutaka Fujii[1,2], Yuki Kako[1], Satoshi Asakura[1,2],
Ichiro Kusumi[1]

1 Department of Psychiatry, Hokkaido University Graduate School of Medicine, Sapporo, Hokkaido, Japan,
2 Health Care Center of Hokkaido University, Sapporo, Hokkaido, Japan

☯ These authors contributed equally to this work.
‡ These authors also contributed equally to this work.
* nmitsui@med.hokudai.ac.jp

## Abstract

### Aim

To enable early identification of university students at high risk for suicide, we examined personality as a predictive factor for major depressive episodes and suicide-related ideation.

### Methods

From 2011 to 2013, we administered the Patient Health Questionnaire-9 (PHQ-9) and the Temperament and Character Inventory (TCI) to 1,997 university students at enrollment (T1). We previously conducted a study using the same data set; this is a re-analysis of the dataset. To prevent contamination of data, participants diagnosed with a depressive episode were excluded at T1. Three years after enrollment (T2), we re-administered the PHQ-9 to the same students. We statistically compared TCI scores at T1 among depressive episode groups and suicide-related ideation groups. Two-way ANOVA and Cochran-Armitage trend tests were used to analyze the relationships between personality traits, depressive episodes, and suicide-related ideation.

### Results

The PHQ-9 summary scores at baseline (T1) were 3.0 (±2.7), with female students scoring 4.6 (±2.9) and male students 2.9 (±2.6, $p = 0.025$). The major depressive episode group at T2 had lower self-directedness (SD) scores at T1 than the non-depressive episode control group. The suicide-related ideation (SI) group at T2 also had higher harm avoidance (HA), lower SD, and lower cooperativeness (C) scores than the non-SI group at T1. The Cochran-Armitage trend tests revealed significant associations between character configurations composed of SD and C, and both depressive episodes at T2 and SI at T2.

**Funding:** This study was supported by the Japan Society for Promotion of Science KAKENHI Grant Number JP18K07583 (SA).

**Competing interests:** KT has received personal fees from Tsumura & Co. and Otsuka Pharmaceutical. NM received lecture fees from Mochida Pharmaceutical. YF has received personal fees from Yoshitomiyakuhin, Otsuka Pharmaceutical, Dainippon Sumitomo Pharma, Eisai and Meiji Seika Pharma. YK has received honoraria from Dainippon Sumitomo Pharma, Eli Lilly, Otsuka Pharmaceutical, Tanabe Mitsubishi Pharma, and Yoshitomiyakuhin. SA has received honoraria from Mochida Pharmaceutica and Yoshitomiyakuhin. IK has received honoraria from Astellas, Daiichi Sankyo, Dainippon Sumitomo Pharma, Eisai, Eli Lilly, Janssen Pharmaceutical, Kyowa Hakko Kirin, Lundbeck, Meiji Seika Pharma, MSD, Mylan, Novartis Pharma, Ono Pharmaceutical, Otsuka Pharmaceutical, Pfizer, Shionogi, Shire, Taisho Toyama Pharmaceutical, Takeda Pharmaceutical, Tanabe Mitsubishi Pharma, Tsumura, and Yoshitomiyakuhin, and has received research/grant support from Astellas, Daiichi Sankyo, Dainippon Sumitomo Pharma, Eisai, Eli Lilly, Kyowa Hakko Kirin, Mochida Pharmaceutical, MSD, Novartis Pharma, Otsuka Pharmaceutical, Pfizer, Shionogi, and Takeda Pharmaceutical. There are no patents, products in development or marketed products to declare. This does not alter our adherence to PLOS ONE policies on sharing data and materials. The other authors do not have any potential competing interests.

## Conclusion

The temperament feature of high HA at baseline and character configurations of low SD and low C at baseline are the most contributory predictors for the novel development of depressive episodes and SI among Japanese university students.

## 1. Introduction

Suicide is a major public health concern [1] and has been the leading cause of death among young adults in Japan during the past decade [2]. Among university students worldwide, the prevalence of depression, a strong risk factor for suicide [3], is reported to be 30.6%–substantially higher than in the general population [4]. However, a recent study revealed that only a small percentage of suicide completers had previously been diagnosed with psychiatric disorders and received services through a university health center [5]. This suggests that early identification and intervention could prevent suicides among high-risk students, prompting the need for studies on the predictors of major depressive episodes (MDEs) and suicide-related ideation (SI, encompassing ideation of both suicide and self-harm). This study uses the term SI for two reasons. First, it is difficult to make a strict distinction between clear suicidal intent and modest suicidal intent in the act of injuring oneself. Second, according to Silverman's definition [6], SI has three subcategories—no suicidal intent, an undetermined degree of suicidal intent, and some suicidal intent—that are almost consistent with the ninth item of Patient Health Questionnaire-9 (PHQ-9).

Several factors have been reported for MDE vulnerability [7]. Previously, prospective studies have shown specific personality traits to be a significant predictor of MDE [8, 9] or SI [10]. An increasing number of studies have used the Temperament and Character Inventory (TCI) to explore personality traits. In cross-sectional studies, patients with MDE showed higher harm avoidance (HA) and lower self-directedness (SD) than healthy participants [11–13]. However, in longitudinal designs for general populations with follow-up periods of less than 1 year to 15 years [14–19], high HA and low SD were significant predictive factors for future depressive symptoms.

In our previous studies on university students [20, 21] using the TCI and PHQ-9, we observed a relationship between MDE and SI, and low SD and low cooperativeness (C). SD relates to self-determination and an individual's ability to control a situation in accordance with their individually chosen goals and values, while C relates to individual differences in how much people identify with and accept other people [22]. In general, SD and C tend to increase with age and low scores are assumed to be an index of immature personality [11, 22]. Thus, when participants are young—for example, university students—character configurations can be of particular importance in the context of suicide prevention.

While some character configurations have been assumed to predict the novel onset of MDE and SI [14–17, 19, 21], causality has yet to be clearly shown, due to heterogeneity at baseline. Although several longitudinal studies have used the TCI, few have controlled for baseline depressive symptoms [14–16, 19, 21]. TCI scales are influenced by current depressive state (known as "state-trait effect") [12, 23]; therefore, if depressive episodes are present, natural personality traits may not be evaluated accurately. To address this concern and to more accurately verify whether character configuration can predict the novel onset of MDE and SI, subjects in a depressive state at baseline should be excluded from the sample to prevent data contamination. This exclusion allows us to focus on the new onset of depressive state.

Based on the background provided above, we conducted a re-analysis of the dataset from a prior study [21], with several methodological differences. Our previous study [20] reported that the prevalence of MDE and SI among university students decreased as the character configuration became more mature. However, because that was a cross-sectional study, the depressive symptoms—which can affect SD and C in the TCI—were not considered. In another previous study [21], we adopted a longitudinal design that enabled us to compare the prevalence of MDE and SI at two timepoints. However, even in this previous report, we did not control for depressive symptoms at baseline. In the present study, we attempted to control for bias related to state-effect by excluding depressive subjects at baseline.

To enable early identification of students at high risk for suicide, this study aimed to elucidate character configurations as a predictive factor of MDEs and SI among university students.

## 2. Material and methods

### 2.1 Participants

Participant flow is shown in Fig 1. As we included the same participants as in our previous study [21], the same procedure was followed. The PHQ-9 and the TCI were administered to students who enrolled at the university in 2011, 2012, and 2013. The number of enrolled students in 2011, 2012, and 2013 were 2,606, 2,600, and 2,591, respectively. Three years later, we administered the PHQ-9 to the same students in April 2014, 2015, and 2016 to detect new onset MDEs and SI. We defined the first year as T1 and the time of retest as T2. We excluded

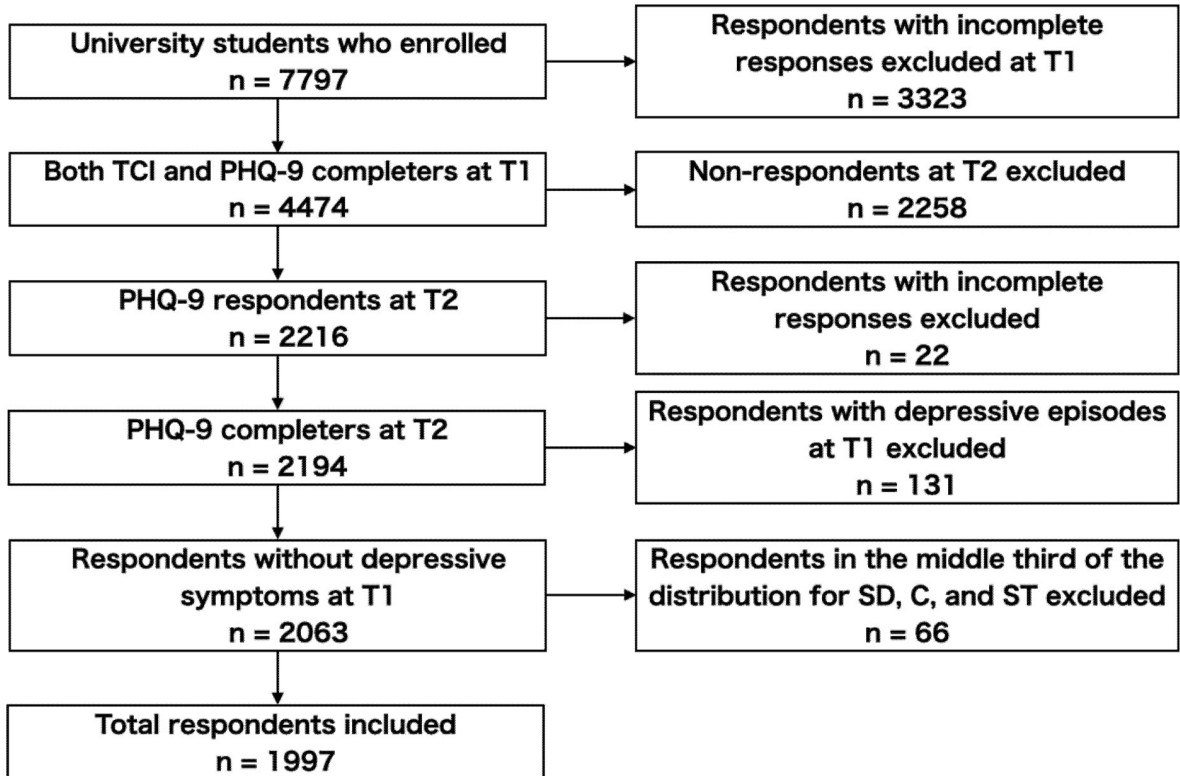

**Fig 1. Participant flow.** C, cooperativeness; PHQ-9, Patient Health Questionnaire-9; SD, self-directedness; SI, suicide-related ideation; ST, self-transcendence; T1, the time students enrolled at university; T2, three years after enrolment; TCI, Temperament and Character Inventory.

all students with incomplete responses to the TCI at T1, the PHQ-9 at T1, and/or the PHQ-9 at T2. The self-rating scales were completed by a total of 2,194 students (28.1% of the total number of students enrolled at Hokkaido University in 2011, 2012, and 2013). The prevalence of MDE at T1 and T2 was 2.0% and 2.3%, respectively. This is consistent with data collected in 2010 [20], where the prevalence was 2.9% and indicated a low selection bias between T1 and T2. We excluded 131 participants with other depressive episodes (ODE) and MDE at T1. To evaluate character configurations, we categorized the non-depressive students who completed all tests ($N$ = 2,063) as above and below the median for each of the three-character traits—SD, C, and self-transcendence (ST)—after excluding 67 participants who were in the middle third of the distribution for all three traits. Ultimately, data from 1,997 students were used in the analysis.

Written informed consent was obtained from all participants prior to completion of the TCI and PHQ-9 at T1 and the PHQ-9 at T2. This study was conducted in accordance with the ethical standards established by the 1964 Declaration of Helsinki (amended in Fortaleza, October 2013) and approved by the Ethical Committee of Hokkaido University Graduate School of Medicine (approval number: 12–00).

## 2.2 Measures

In this study, we used the PHQ-9 to assess depressive symptoms and the TCI to assess personality traits. The tests were administered in a mark-sheet written format. The information obtained from the questionnaires was not disclosed and was managed appropriately at the healthcare center of Hokkaido University. If severe depressive state or SI were detected, the healthcare center contacted the students telephonically or via email and started intervention, as these students are considered to be at high risk of suicide.

**2.2.1 Patient health Questionnaire-9.** The PHQ-9 is a self-report questionnaire for the Primary Care Evaluation of Mental Disorders (PRIME-MD). The validity of the PHQ-9 as a screening tool for MDE has been confirmed in primary care settings and two meta-analyses revealed its high sensitivity (0.80; [95% CI: 0.71–0.87] and 0.77 [95% CI: 0.71, 0.84], respectively) and specificity (0.92 [95% CI: 0.88–0.95] and 0.94 [95% CI: 0.90, 0.97], respectively) [24, 25]. We used the Japanese version of the PHQ-9, which has high validity in primary care [26] and psychiatric settings [27]. The PHQ-9 can diagnose the severity of depression by a summary score and depressive episodes by a diagnostic algorithm [28] that distinguishes between MDE and ODE. PHQ-9 summary scores were divided into five categories: 0–4, 5–9, 10–14, 15–19, and $\geq$ 20 [29], with depressive symptoms defined as a PHQ-9 summary score of $\geq$ 5. It should be noted that the term "MDE" should be interpreted with caution, as it is a classification based on the PHQ-9 algorithm screening tool, and not a diagnosis that has been determined through a structured interview. As this study aimed to analyze the relationships in the data obtained from the screening, we did not conduct structured interviews with a reduced sample size.

When participants indicated having "thoughts that [they] would be better off dead or of hurting [themselves] in some way" at least "several days" out of the week, they were regarded as having "suicide-related ideation," an encompassing ideation of both suicide and self-harm and coinciding with the ninth item of the PHQ-9 score $\geq$ 1.

**2.2.2 Temperament and character inventory.** The TCI is a self-rating scale of personality based on the psychological model of personality proposed by Cloninger. [22] The TCI assesses seven dimensions of personality divided into four temperament and three character dimensions. The four temperament dimensions are novelty seeking (NS), harm avoidance (HA), reward dependence (RD), and persistence (P), while the three-character dimensions comprise

SD, C, and ST. Moreover, possible character configurations are defined by high and low scores on the three-character dimensions of the TCI [11]. In this study, we used the 125-item Japanese version of the TCI with a four-point answer scale. Kijima et al. [30] reported that a four-point scale was superior to a dichotomous scale in terms of internal consistency, as expressed by the Cronbach's α coefficient. The Japanese version of the TCI has been confirmed as a valid and reliable measure of temperament and character among university students [31].

### 2.3 Statistical analyses

Demographic data were compared between female and male participants using *t*-tests and chi-squared tests. A two-way ANOVA was used to compare TCI scores among the non-depressive control (NC), ODE, and MDE groups. Diagnoses and sex were used as two categorical factors for the analyses, because sex differences have been reported for the TCI [12]. Tukey's honestly significant difference (HSD) test was applied as a post-hoc analysis. Seven independent factors, namely the NS, HA, RD, P, SD, C, and ST scores from the TCI, were used to assess future risk of MDE and SI. Next, we analyzed the association between character configurations, depressive episodes, and SI. To confirm the order of character dimensions, a logistic regression analysis was performed. Thereafter, Cochran-Armitage trend tests were conducted to show the trend between character configurations and the development of depressive episodes and SI at T2. All tests were two-tailed and differences were considered significant at $p < 0.001$ to compensate for the effects of the large sample size and multiple comparisons. JMP Pro software version 14.0 (SAS Institute Inc., Cary, NC, USA) was used for the analyses.

## 3. Results

### 3.1 Baseline data

Female participants comprised 723 (36.2%) of the 1,997 participants. The PHQ-9 diagnostic algorithm was used to divide participants into three groups—NC, ODE, and MDE—based on their scores at T2. The participants were also divided into two groups according to the presence or absence of SI at T2. The demographic data of depressive episode groups and SI groups are shown in Table 1. Sex differences were not observed among NC, ODE, and MDE groups ($\chi^2 = 5.51$, $p = 0.06$), or among non-SI and SI groups ($\chi^2 = 1.50$, $p = 0.22$). The PHQ-9 summary scores of ODE and MDE were significantly higher than those of NC (ANOVA F = 27.8, $p < 0.001$; vs ODE, F = 11.1, $p < 0.001$; vs MDE, F = 16.5, $p < 0.001$). The prevalence of SI differed significantly among NC, ODE, and MDE groups (All, $\chi^2 = 23.5$, $p < 0.001$; female, $\chi^2 = 9.86$, $p = 0.007$, male, $\chi^2 = 14.1$, $p < 0.001$), and among non-SI, and SI groups (All, $\chi^2 = 37.3$, $p < 0.001$; female, $\chi^2 = 27.1$, $p < 0.001$, male, $\chi^2 = 12.4$, $p < 0.001$).

### 3.2 Personality traits of depressive episodes groups

The mean TCI scores of depressive episodes groups by sex are shown in Table 2. To determine interaction effects between sex and depressive episodes, we performed two-way ANOVA tests (sex × depressive episodes). The two-way ANOVA revealed significant effects of depressive episode at T2 on SD scores (F[2, 1993] = 20.26, $p < 0.001$). The Tukey's HSD tests were then performed as a post-hoc analysis. The MDE group had significantly lower scores on SD ($p < 0.001$) than the NC group. The ODE group had significantly higher scores than the NC group on SD only. Sex differences were also seen in RD (F[1, 1993] = 77.94, $p < 0.001$) and SD (F[1, 1993] = 43.81, $p < 0.001$). However, no interaction effects were observed between sex and depressive episode on each TCI score (Table 3).

**Table 1. Baseline data.**

|  | Depressive episodes at T2 | | | Suicide-related ideation at T2 | |
|---|---|---|---|---|---|
|  | NC | ODE | MDE | Non-SI | SI |
| $N^{\dagger}$ | | | | | |
| All | 1900 | 67 | 30 | 1897 | 100 |
| Female (%) | 680 (35.8%) | 26 (38.8%) | 17 (56.7%) | 681 (35.9%) | 42 (42.0%) |
| Male (%) | 1220 (64.2%) | 41 (61.2%) | 13 (43.3%) | 1216 (64.1%) | 58 (58.0%) |
| Age$^{\ddagger}$ | | | | | |
| All | 19.2 (0.9) | 19.2 (0.8) | 19.1 (0.7) | 19.2 (0.9) | 19.4 (1.8) |
| Female | 19.2 (0.9) | 19.3 (0.8) | 18.8 (0.8) | 19.2 (0.9) | 19.1 (0.7) |
| Male | 19.2 (1.0) | 19.2 (0.8) | 19.3 (0.8) | 19.2 (0.8) | 19.6 (2.3) |
| PHQ-9 summary score$^{\ddagger}$ | | | | | |
| All | 2.9 (2.6) | 4.6* (2.9) | 5.4* (3.9) | 2.9 (2.6) | 5.3* (3.0) |
| Female | 3.0 (2.6) | 4.6* (2.8) | 5.3* (3.9) | 3.1 (2.6) | 4.9* (3.1) |
| Male | 2.8 (2.7) | 4.5* (3.0) | 5.9* (3.9) | 2.8 (2.7) | 5.6* (2.9) |
| SI (%)$^{\dagger}$ | | | | | |
| All | 55 (2.9%) | 4 (6.0%) | 8** (26.7%) | 49 (2.6%) | 18* (18.0%) |
| Female | 20 (2.9%) | 1 (3.9%) | 4** (23.5%) | 15 (2.2%) | 10* (23.8%) |
| Male | 35 (2.9%) | 3 (7.3%) | 4** (30.8%) | 34 (2.8%) | 8* (13.8%) |

$^{\dagger}$ = chi-squared test;

$^{\ddagger}$ = ANOVA or Tukey's HSD test for depressive episode at T2, and *t*-test for suicide-related ideation at T2.* $p < 0.001$ (compared with NC),

** $p < 0.001$.

Abbreviations: MDE, Major Depressive Episode; NC, Non-depressive Control; ODE, Other Depressive Episode; PHQ-9, Patient Health Questionnaire-9; SI, suicide-related ideation.

## 3.3 Personality traits of SI group

The two-way ANOVA scores were adapted for analyzing the TCI score between non-SI and SI. The mean TCI scores of SI groups by sex are shown in Table 4. To measure the interaction effects between sex and SI groups, we performed two-way ANOVA tests (sex × SI). The two-way ANOVA revealed significant effects of depressive episode at T2 on HA scores

**Table 2. Baseline TCI scores among NC, ODE, and MDE.**

|  | NC | | | ODE | | | MDE | | |
|---|---|---|---|---|---|---|---|---|---|
|  | All | Female | Male | All | Female | Male | All | Female | Male |
|  | $N = 1900$ | $N = 680$ | $N = 1220$ | $N = 67$ | $N = 26$ | $N = 41$ | $N = 30$ | $N = 17$ | $N = 13$ |
| Temperament | | | | | | | | | |
| NS | 28.6 (6.4) | 28.5 (6.6) | 28.7 (6.3) | 29.5 (8.0) | 30.9 (8.5) | 28.7 (7.7) | 30.4 (7.1) | 28.4 (5.4) | 33.0 (8.4) |
| HA | 33.9 (9.1) | 33.2 (9.4) | 34.3 (9.0) | 35.5 (10.5) | 34.3 (10.7) | 36.3 (10.4) | 39.1 (9.8) | 40.8 (8.2) | 37.0 (11.7) |
| RD | 26.4 (5.8) | 28.0 (5.8) | 25.6 (5.8) | 24.8 (6.6) | 25.8 (5.5) | 24.2 (7.3) | 24.0 (5.8) | 24.5 (3.6) | 23.2 (7.9) |
| P | 8.3 (2.7) | 8.7 (2.7) | 8.1 (2.7) | 8.3 (3.2) | 8.4 (3.3) | 8.2 (3.1) | 8.1 (3.3) | 8.3 (2.8) | 7.8 (4.0) |
| Character | | | | | | | | | |
| SD | 44.2 (9.1) | 46.0 (8.1) | 43.3 (9.2) | 39.5 (8.9) | 42.5 (8.1) | 37.5 (9.0) | 36.9 (8.3) | 38.7 (7.2) | 34.5 (9.3) |
| C | 48.7 (7.9) | 50.8 (7.2) | 47.5 (8.0) | 47.1 (8.6) | 48.3 (7.3) | 46.3 (9.3) | 43.9 (8.2) | 47.8 (6.5) | 38.8 (7.4) |
| ST | 14.1 (6.8) | 14.7 (6.5) | 13.8 (7.0) | 15.5 (7.4) | 17.5 (7.5) | 14.1 (7.1) | 14.5 (8.0) | 15.4 (7.9) | 13.5 (8.2) |

Abbreviations: C, cooperativeness; HA, harm avoidance; MDE, Major Depressive Episode; NC, Non-depressive Control; NS, novelty seeking; ODE, Other Depressive Episode; P, persistence; PHQ-9, Patient Health Questionnaire-9; RD, reward dependence; SD, self-directedness; ST, self-transcendence.

**Table 3. Comparison of TCI scores using a two-way ANOVA among NC, ODE, and MDE.**

|  | Source of variations | F statistics | p | Tukey HSD test | | |
|--|--|--|--|--|--|--|
|  |  |  |  | NC vs. ODE | ODE vs. MDE | NC vs. MDE |
| NS | Sex | 0.340 | 0.560 |  |  |  |
|  | Diagnosis | 1.665 | 0.190 |  |  |  |
|  | Sex × diagnosis | 2.862 | 0.057 |  |  |  |
| HA | Sex | 7.044 | 0.008 |  |  |  |
|  | Diagnosis | 6.104 | 0.002 | 0.351 | 0.169 | 0.006 |
|  | Sex × diagnosis | 1.129 | 0.324 |  |  |  |
| RD | Sex | 77.94 | < 0.001 | (Female > Male) |  |  |
|  | Diagnosis | 6.419 | 0.002 | 0.059 | 0.075 | 0.779 |
|  | Sex × diagnosis | 0.315 | 0.730 |  |  |  |
| P | Sex | 21.96 | < 0.001 | (Female > Male) |  |  |
|  | Diagnosis | 0.302 | 0.740 |  |  |  |
|  | Sex × diagnosis | 0.188 | 0.828 |  |  |  |
| SD | Sex | 43.81 | < 0.001 | (Female > Male) |  |  |
|  | Diagnosis | 20.26 | < 0.001 | < 0.001 ↓ | 0.407 | < 0.001 ↓ |
|  | Sex × diagnosis | 0.607 | 0.545 |  |  |  |
| C | Sex | 83.45 | < 0.001 | (Female > Male) |  |  |
|  | Diagnosis | 8.815 | 0.002 | 0.230 | 0.157 | 0.003 |
|  | Sex × diagnosis | 2.258 | 0.105 |  |  |  |
| ST | Sex | 9.507 | 0.002 |  |  |  |
|  | Diagnosis | 1.194 | 0.303 |  |  |  |
|  | Sex × diagnosis | 1.105 | 0.331 |  |  |  |

Abbreviations: C, cooperativeness; HA, harm avoidance; MDE, Major Depressive Episode; NC, Non-depressive Control; NS, novelty seeking; ODE, Other Depressive Episode; P, persistence; RD, reward dependence; SD, self-directedness; ST, self-transcendence.

[↑], higher in ODE or MDE group than in NC group.

[↓], lower in ODE or MDE group than in NC group.

**Table 4. Baseline TCI scores among non-SI and SI.**

|  | Non-SI | | | SI | | |
|--|--|--|--|--|--|--|
|  | All | Female | Male | All | Female | Male |
|  | N = 1897 | N = 681 | N = 1216 | N = 100 | N = 42 | N = 58 |
| Temperament |  |  |  |  |  |  |
| NS | 28.6 (6.5) | 28.5 (6.7) | 28.6 (6.4) | 30.6 (6.9) | 29.9 (6.6) | 31.1 (7.0) |
| HA | 33.9 (9.1) | 33.2 (9.4) | 34.3 (8.9) | 37.0 (10.8) | 36.2 (10.7) | 37.5 (10.9) |
| RD | 26.4 (5.9) | 27.8 (5.8) | 25.6 (5.7) | 25.0 (6.3) | 27.5 (5.5) | 23.2 (6.3) |
| P | 8.3 (2.7) | 8.7 (2.7) | 8.1 (2.7) | 7.8 (3.1) | 8.5 (2.9) | 7.3 (3.2) |
| Character |  |  |  |  |  |  |
| SD | 44.2 (9.1) | 46.0 (8.7) | 43.2 (9.2) | 39.4 (9.5) | 41.3 (8.7) | 37.9 (9.9) |
| C | 48.7 (7.8) | 50.7 (7.1) | 47.6 (8.0) | 45.3 (8.6) | 49.1 (8.4) | 42.5 (7.7) |
| ST | 14.2 (6.8) | 14.7 (6.4) | 13.9 (7.0) | 14.6 (7.7) | 17.0 (8.0) | 12.7 (7.1) |

Abbreviations: C, cooperativeness; HA, harm avoidance; NS, novelty seeking; P, persistence; RD, reward dependence; SD, self-directedness; SI, suicide-related ideation; ST, self-transcendence.

**Table 5. Comparison of TCI scores using a two-way ANOVA among non-SI and SI.**

|    | Source of variations | F statistics | *p* |
|----|----------------------|--------------|-----|
| NS | Sex | 0.335 | 0.563 |
|    | SI | 8.933 | 0.003 |
|    | Sex × SI | 0.547 | 0.460 |
| HA | Sex | 6.578 | 0.010 |
|    | SI | 11.08 | < 0.001 ↑ |
|    | Sex × SI | 0.01 | 0.917 |
| RD | Sex | 76.20 | < 0.001 (Female > Male) |
|    | SI | 6.761 | 0.009 |
|    | Sex × SI | 2.929 | 0.087 |
| P  | Sex | 22.19 | < 0.001 (Female > Male) |
|    | SI | 4.091 | 0.043 |
|    | Sex × SI | 1.524 | 0.217 |
| SD | Sex | 41.80 | < 0.001 (Female > Male) |
|    | SI | 29.33 | < 0.001 ↓ |
|    | Sex × SI | 0.152 | 0.697 |
| C  | Sex | 82.14 | < 0.001 (Female > Male) |
|    | SI | 21.44 | < 0.001 ↓ |
|    | Sex × SI | 4.791 | 0.029 |
| ST | Sex | 9.589 | 0.002 |
|    | SI | 0.226 | 0.635 |
|    | Sex × SI | 6.089 | 0.014 |

Abbreviations: C, cooperativeness; HA, harm avoidance; MDE, Major Depressive Episode; NC, Non-depressive Control; NS, novelty seeking; ODE, Other Depressive Episode; P, persistence; RD, reward dependence; SD, self-directedness; SI, suicide-related ideation; ST, self-transcendence.

[↑], higher in SI group than in non-SI group.

[↓], lower in SI group than in non-SI group.

(F[1, 1994] = 11.08, $p < 0.001$), SD scores (F[1, 1994] = 29.33, $p < 0.001$), and C scores (F[1, 1994] = 21.44, $p < 0.001$). Sex differences were also seen in RD (F[1, 1993] = 76.20, $p < 0.001$), P (F[1, 1993] = 22.19, $p < 0.001$), SD (F[1, 1993] = 41.80, $p < 0.001$) and C (F[1, 1993] = 82.14, $p < 0.001$). However, no interaction effects were observed between sex and SI group on each TCI score (Table 5).

### 3.4 Character configurations and vulnerability

To confirm the contributions of personality traits to the development of both depressive episodes (ODE and MDE) and SI, we performed a logistic regression analysis with SD, C, and ST as independent variables. Results of this analysis indicated that SD was the most contributory factor (for depressive episode, $\chi^2 = 33.1$, OR = 0.94, $p < 0.001$; for SI, $\chi^2 = 26.1$, OR = 0.94, $p < 0.001$) followed by C (for depressive episode, $\chi^2 = 9.86$, OR = 0.96, $p = 0.002$; for SI, $\chi^2 = 18.1$, OR = 0.95, $p < 0.001$), however ST did not significantly contribute (for depressive episode, $\chi^2 = 2.17$, OR = 1.02, $p = 0.141$; for SI, $\chi^2 = 0.31$, OR = 1.01, $p = 0.575$) to the development of both depressive episode and SI.

Based on this tendency, we compared the prevalence of depressive episodes at T2 among four categories of possible combinations: low SD/low C (sc), low SD/high C (sC), high SD/low C (Sc), and high SD/high C (SC). The number of sc, sC, Sc, and SC were 594, 350, 385, and 668, respectively. The prevalence of depressive episodes in sc, sC, Sc, and SC were 7.91%,

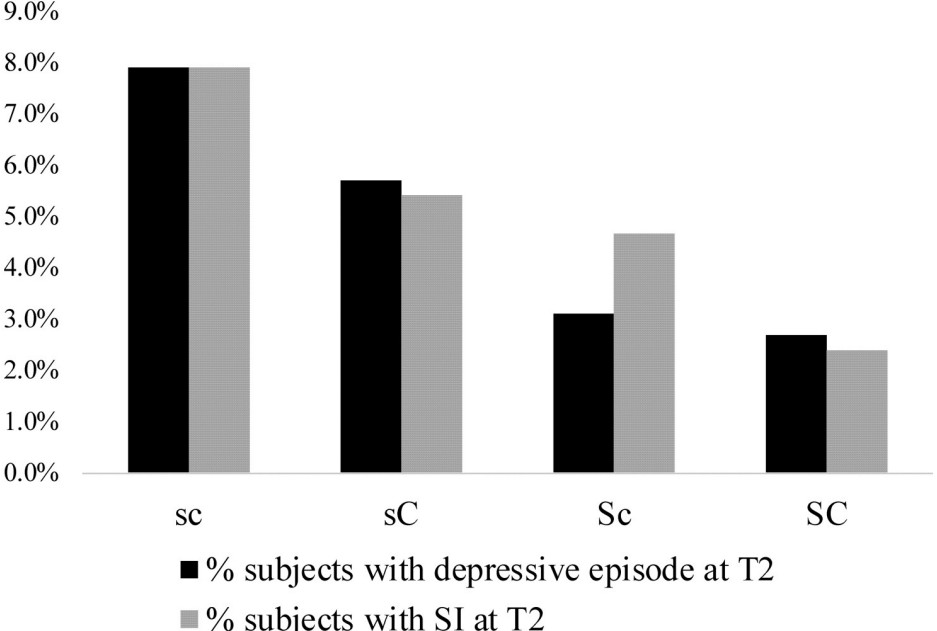

**Fig 2. Cochran-Armitage trend test of four character configurations.** Cochran-Armitage trend test: Depressive Episode, $\chi^2_{trend}$ = 20.7, $p < 0.001$, $\chi^2_{liniarity}$ = 0.87, $p$ = 0.83; SI, $\chi^2_{trend}$ = 19.9, $p < 0.001$, $\chi^2_{liniarity}$ = 1.11, $p$ = 0.77. C, cooperativeness score higher than median; c, cooperativeness score lower than median; S, self-directedness score higher than median; s, self-directedness score lower than median; SI, suicide-related ideation; T2, three years after enrolment.

5.71%, 3.12%, and 2.69%, while that of SI in sc, sC, Sc, and SC were 7.91%, 5.43%, 4.68%, and 2.40%, respectively. The prevalence of depressive episodes and SI among the four types of character configurations are shown in Fig 2. The Cochran-Armitage trend tests revealed that the prevalence of depressive episodes at T2 decreased when SD and/or C were high at T1; that is, character profiles became mature ($\chi^2_{trend}$ = 20.7, $p < 0.001$); the same tendency was observed in subjects with SI ($\chi^2_{trend}$ = 19.9, $p < 0.001$).

Lastly, the sensitivity, specificity, and positive predictive value for depressive episode when using the data of SC and sc groups were 0.723, 0.543, and 0.079 respectively. Similarly, the sensitivity, specificity, and positive predictive value for SI when using the data of SC and sc groups were 0.746, 0.544, and 0.079 respectively.

## 4. Discussion

The main finding of this study is that the character configurations of low SD and low C at university enrollment (T1) could be a predictive factor for the novel development of depressive episodes and SI, three years later (T2). In the temperament dimension, a high HA score could be a predictive factor for the novel development of depressive episodes and SI. In this study, we were able to accurately measure the TCI score at baseline, because we controlled for bias of state effects by excluding subjects with other depressive episodes and major depressive episode at T1 (i.e., baseline). It should be noted that the present study is a re-analysis study that uses part of a dataset from our previous our study [21]; however, the analysis method is completely different and we use a method in this study that could not be applied in our previous study [21]. Suicide is the leading cause of death in Japan among young people (from adolescence to people in their 30s, including university students). This high occurrence makes it an extremely

important issue, compared with other regions. Moreover, the issue of suicide among young people is becoming increasingly important, especially during the current COVID-19 pandemic. Therefore, it is meaningful to conduct a re-analysis from a different perspective.

The present results coincide with those of previous reports. In previous cross-sectional and longitudinal studies in Japan, high SD and high C scores were substantial protective factors against future depressive episodes [17, 18, 20, 21]. In longitudinal studies in particular, low SD scores were reported to predict depressive symptoms among Japanese college students [2, 17, 18]. Furthermore, previous longitudinal studies conducted in the United States and in Europe [13, 14, 16, 19] suggest that the association between low SD and major depressive episode or SI was non-specific in a Japanese population. In light of these reports, low SD is considered to be a significant predictor of future depressive episodes and SI among newly-enrolled university students. Conversely, high SD is considered to be a protective factor against the future development of depressive episodes. The same holds relatively true for C. Hence, the character configurations of low SD and low C indicate vulnerability to MDEs and SI among university students.

Regarding the temperament dimension, this study observed higher HA score in MDE and SI groups. As reported by several previous studies [11–13], high HA has been thought to be another predictive factor for future depressive episodes and SI among university students. HA features tendencies such as inhibiting one's behavior, which is related to anticipatory worry, fear of uncertainty, shyness, and rapid fatigability [22]; this makes HA one of the most important factors for vulnerability to MDE and SI. The other temperament dimensions (i.e., NS, RD, and P) did not have a clear relationship with future depressive episodes or SI.

Cloninger [11] distinguished character types in terms of their character dimension combinations and reported the association between those types and depression and suicide attempts. In the current study, the combination of low SD and low C was the most significant predictor of both future depressive episodes and SI. Cloninger and Zohar [32] and Josefsson [33] reported that SD and C were associated with various aspects of well-being in non-clinical adult samples [32, 33]. Garcia et al. [34] reported that SD was related to subjective well-being in adolescents and young adults. These findings imply that high SD and high C are associated with resilience in dealing with certain types of stress and can be a protective factor for stress-related mental disorders. By its nature, high SD and high C may contribute as protective factors for suicide by facilitating transformation to more adaptive behaviors and formation of a social network.

In general, SD and C tend to increase with age and low scores are assumed to be an index of immature personality [11, 22, 35]. SD measures self-determination and an individual's ability to control a situation in accordance with their individually chosen goals and values [22]. Cooperativeness measures an individual's social tolerance, empathy, helpfulness, and compassion [22]. These character dimensions could vary among young people such as university students, and low scores on both SD and C could affect their social adaptation. The assumption that character configurations may change over time is worth considering in the context of suicide prevention. Interventions to enhance students' SD and C at university enrollment may prevent depressive episodes and suicide attempts and increase their resilience and well-being; such interventions have not yet been implemented.

It is worth noting that our study included a three-year follow-up, adopted non-clinical adult participants, and had a big sample size, with close to two thousand participants. Clinically, we observed no meaningful difference between NC, ODE, and MDE on the PHQ-9 summary scores at T1 because of the low scores in all groups, even though those in MDE and ODE were statistically significantly higher than those in NC. By adopting a longitudinal design and preventing contamination of data related to depressive symptoms at T1, we were able to

evaluate the association between character configurations and the new onset of depressive episodes or SI.

Several limitations similar to those noted in our previous report [21] need to be considered when interpreting our results. First, we used a self-report questionnaire, the PHQ-9, to assess depressive episodes and SI. Structured diagnostic interviews with each participating student would improve diagnostic accuracy; however, it is difficult to use interviews in a large-scale study such as ours. Second, each participant was assessed at only two points in time, and we could not evaluate whether depressive symptoms persisted between those time points [36]. Because the PHQ-9 considers the last two weeks, we should especially note the possibility that depressive symptoms and SI might have waxed and waned over the course of three years, leading to the absence of symptoms at the time of evaluation. However, additional points of assessment would likely be troublesome for participants and increase the dropout rate. Indeed, the present study, which assessed participants at only two points, had an overall low participation rate, considering the number at enrollment. Third, there are sampling issues that may have affected our results. Only 28% of the university enrollments participated in this study. Further, as we did not use random or systematic sampling, there could possibly be sampling biases. However, the main purpose of administering the PHQ-9 and TCI was to screen for mental health. After explaining the structure and purpose of the research before the tests were conducted, we asked as many students as possible to participate voluntarily. Even though it is desirable to have as many participants as possible, the number of respondents decreased to 28%, due to the research design that required multiple questionnaires and the time interval between the tests being as long as three years. Continuous efforts should be made in future to increase the response rate. Finally, this study did not distinguish between suicide ideation and self-harm. We evaluated suicide and self-harm ideation using Item 9 of the PHQ-9; however, the relationship between suicide and self-harm behavior in the university population is unclear, as more than half of the students who had engaged in self-harm behavior reported never having considered or attempted suicide [37]. Therefore, ideas of self-harm are not always related to SI.

In conclusion, the character configuration of low SD and low cooperativeness is one of the most contributory predictive factors for novel development of depressive episodes and SI among Japanese university students. In the temperament dimension, high HA is also one of the most important predictive factors for depressive episode and SI. Character profiles will have a strong impact on future development of major depressive episodes and SI among Japanese university students.

## Acknowledgments

We are grateful to Dr. Satoshi Hashino, director of the Health Care Center of Hokkaido University, for supporting this study; Dr. Takeshi Inoue, Professor at the Department of Psychiatry, Tokyo Medical University, for providing valuable suggestions and comments to improve the study; and Hiroko Takeda, Choichiro Saito, Rui Kawashima, and Kaai Ishihara, staff members at the Health Care Center of Hokkaido University, for the assistance with data collection. We would like to thank Editage (www.editage.com) for English language editing.

## Author Contributions

**Conceptualization:** Keisuke Takanobu, Nobuyuki Mitsui.

**Data curation:** Keisuke Takanobu, Nobuyuki Mitsui, Satoshi Asakura.

**Formal analysis:** Keisuke Takanobu, Nobuyuki Mitsui.

**Funding acquisition:** Satoshi Asakura.

**Investigation:** Keisuke Takanobu, Nobuyuki Mitsui.

**Methodology:** Keisuke Takanobu, Nobuyuki Mitsui, Satoshi Asakura.

**Project administration:** Nobuyuki Mitsui.

**Resources:** Nobuyuki Mitsui.

**Software:** Nobuyuki Mitsui.

**Supervision:** Shinya Watanabe, Kuniyoshi Toyoshima, Yutaka Fujii, Yuki Kako, Satoshi Asakura, Ichiro Kusumi.

**Validation:** Nobuyuki Mitsui.

**Visualization:** Keisuke Takanobu.

**Writing – original draft:** Keisuke Takanobu.

**Writing – review & editing:** Shinya Watanabe, Kuniyoshi Toyoshima, Yutaka Fujii, Yuki Kako, Satoshi Asakura, Ichiro Kusumi.

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
