## [Decision Letter · Decision Letter 0]

23 Feb 2021

PONE-D-21-04425

Character configuration, major depressive episodes, and suicide-related ideation among Japanese undergraduates

PLOS ONE

Dear Dr. Mitsui,

Thank you for submitting your manuscript to PLOS ONE. After careful consideration, we feel that it has merit but does not fully meet PLOS ONE’s publication criteria as it currently stands. Therefore, we invite you to submit a revised version of the manuscript that addresses the points raised during the review process. We are concerned that you did not make clear to the reader that this paper is a reanalysis of the same data that you have previously published with only a minor adjustment of the sample to exclude those with depressive symptoms at T1.  Your findings are sound and consistent with extensive prior work. The exclusion of the modest number of students with depression initially is a valuable refinement, but it does not change the overall findings.  You will need to cite not only the prior publication in Comprehensive Psychiatry and also that in PLOS ONE (PLoS ONE 13 (7): e0201047. https://doi.org/10.1371/journal.pone.0201047).  There are additional issues and recommendations specified clearly in the attached review, so I am returning this to you for major revision in case you want us to consider it further for possible publication. I welcome that revision but of course I cannot be sure that your revision will be adequate until it is reviewed again.

We look forward to receiving your revised manuscript.

Kind regards,

C. Robert Cloninger, MD, PhD

Academic Editor

PLOS ONE

Journal Requirements:

2) Thank you for stating the following in the Competing Interests section:

[Keisuke Takanobu received personal fees from Tsumura & Co. and Otsuka

Pharmaceutical. Nobuyuki Mitsui received lecture fees from Mochida Pharmaceutical.

Yutaka Fujii received personal fees from Yoshitomiyakuhin, Otsuka Pharmaceutical,

Dainippon Sumitomo Pharma, Eisai and Meiji Seika Pharma. Yuki Kako has received

honoraria from Dainippon Sumitomo Pharma, Eli Lilly, Otsuka Pharmaceutical, Tanabe

Mitsubishi Pharma, and Yoshitomiyakuhin. Satoshi Asakura has received honoraria

from Mochida Pharmaceutica and Yoshitomiyakuhin. Ichiro Kusumi has received

honoraria from Astellas, Daiichi Sankyo, Dainippon Sumitomo Pharma, Eisai, Eli Lilly,

Janssen Pharmaceutical, Kyowa Hakko Kirin, Lundbeck, Meiji Seika Pharma, MSD,

Mylan, Novartis Pharma, Ono Pharmaceutical, Otsuka Pharmaceutical, Pfizer,

Shionogi, Shire, Taisho Toyama Pharmaceutical, Takeda Pharmaceutical, Tanabe

Mitsubishi Pharma, Tsumura, and Yoshitomiyakuhin, and has received research/grant

support from Astellas, Daiichi Sankyo, Dainippon Sumitomo Pharma, Eisai, Eli Lilly,

Kyowa Hakko Kirin, Mochida Pharmaceutical, MSD, Novartis Pharma, Otsuka

Pharmaceutical, Pfizer, Shionogi, and Takeda Pharmaceutical. The other authors do

not have any potential competing interests.].

3)  We noticed you have some minor occurrence of overlapping text with the following previous publication(s), which needs to be addressed:

- https://doi.org/10.1371/journal.pone.0201047

In your revision ensure you cite all your sources (including your own works), and quote or rephrase any duplicated text outside the methods section. Further consideration is dependent on these concerns being addressed.

Reviewers' comments:

Reviewer's Responses to Questions

**Comments to the Author**

1. Is the manuscript technically sound, and do the data support the conclusions?

Reviewer #1: Yes

2. Has the statistical analysis been performed appropriately and rigorously? 

Reviewer #1: Yes

3. Have the authors made all data underlying the findings in their manuscript fully available?

Reviewer #1: Yes

4. Is the manuscript presented in an intelligible fashion and written in standard English?

Reviewer #1: Yes

5. Review Comments to the Author

Reviewer #1: This article describes the findings of data collected from university students in Japan annually over 3 years using the Patient Health Questionnaire-9 (PHQ-9) to measure depression and suicidal ideation and the Temperament and Character Inventory (TCI) to measure personality characteristics. The main aim was to provide a means of early identification of students at risk for major depressive disorder and suicidal ideation. The manuscript reported that low self-directedness (SD), low cooperativeness (C), and high harm avoidance (HA) at baseline were significant predictors of subsequent depression and suicidal ideation. This is a well-written manuscript with good English language usage.

The manuscript noted that numerous other studies have previously reported that low SD, low C, and high HA have been found to be predictive of future depressive episodes and suicidal ideation in university students, including findings from this research team’s prior publications. So what is new and previously unpublished in the overall findings described in this manuscript? It appears that this manuscript simply reports the same findings as the prior articles published from this study with the adjustment of the exclusion of students with depression detected at baseline. It therefore does not seem like this article warrants such lengthy text, as it seems to be merely a replication of prior findings with a repeat analysis of the same data with a slight modification of the sample (6% excluded). The manuscript needs to clarify this clearly front and center in the Introduction, Discussion, and Abstract sections, stating that this study represents a re-analysis of the prior study with this one small methodological difference, and stating why this might be an important analysis to conduct. Were the prior findings simply replicated, and did any new results emerge from this methodological adjustment?

There are some methodological issues that need attention. First, only 28% of the university students participated in this study. How were they selected, or did they simply represent a volunteer or convenience sample? Was there some systematic means of recruitment of this 28% of the student population? A related issue is that if the sample recruitment was not randomly or systematically selected, there could be important sampling biases. The limitations part of the Discussion needs to include comments about this issue and how the authors think sampling issues might have affected the study results.

Also, how were the research instruments administered to the students? Were they confidentially administered? For students reporting current suicidal ideation, how did the study manage their clinical risk?

Another important methodological issue is the use of a self-report symptom scale. The manuscript mentions the excessive burden of instead using a fully diagnostic instrument, especially with such a large sample. Is such a large sample needed, and could studies be designed using smaller sample sizes determined by power estimates that might allow application of diagnostic tools to yield sufficient analysis? It is unfortunate that the Methods and Results sections characterize the PHQ-9 as “diagnostic,” even though it is acknowledged in the limitations section that it is not a diagnostic instrument. Therefore, the use of the term major depressive episode (MDE) throughout the manuscript represents misleading terminology, and a different term for the positive PHQ-9 screening result is needed. Additionally, the item assessing suicidal ideation in the PHQ-9 in this study is overly broad and includes thoughts of self harm that may include contemplation of or engagement in behavior not intended as life-threatening, such as non-lethal self-cutting gestures. This is another limitation that should be acknowledged and discussed in terms of how it might have affected the findings.

Finally, given that the main purpose of this study was to provide early identification of students at risk for major depressive disorder and suicidal ideation that could potentially be addressed by clinical interventions, it would seem that there should be some statistics provided about what proportions of students with depression or suicidal ideation were identified, with indicators such as sensitivity, specificity, and positive and negative predictive values that can confer concrete clinical utility for these purposes.

6. PLOS authors have the option to publish the peer review history of their article (what does this mean?). If published, this will include your full peer review and any attached files.

Reviewer #1: No

---

## [Author Response · Author response to Decision Letter 0]

17 Apr 2021

Dear Editor:

Thank you for the thoughtful comments from yourself and the reviewer regarding our manuscript (PONE-D-21-04425), submitted on February 23, 2021. After reviewing the comments from the editor and the reviewer carefully, we have made the corrections described below, which we hope will meet with your approval:

Responses to Academic Editor:

Thank you for your valuable comments and for coordinating highly suggestive reviews for our manuscripts.

Responses to Reviewer #1:

Comment 1: The manuscript noted that numerous other studies have previously reported that low SD, low C, and high HA have been found to be predictive of future depressive episodes and suicidal ideation in university students, including findings from this research team’s prior publications. So what is new and previously unpublished in the overall findings described in this manuscript? It appears that this manuscript simply reports the same findings as the prior articles published from this study with the adjustment of the exclusion of students with depression detected at baseline. It therefore does not seem like this article warrants such lengthy text, as it seems to be merely a replication of prior findings with a repeat analysis of the same data with a slight modification of the sample (6% excluded). The manuscript needs to clarify this clearly front and center in the Introduction, Discussion, and Abstract sections, stating that this study represents a re-analysis of the prior study with this one small methodological difference, and stating why this might be an important analysis to conduct. Were the prior findings simply replicated, and did any new results emerge from this methodological adjustment?

Response: Thank you for your valuable input on this matter. Our previous study (Temperament and character profiles of Japanese university students with depressive episodes and ideas of suicide or self-harm: A PHQ-9 screening study. http://dx.doi.org/10.1016/j.comppsych.2013.05.014) reported that the prevalence of major depressive episodes (MDE) and suicide-related ideation (SI) among university students decreased as their character configuration became more mature. However, this previous study was a cross-sectional study and we did not control for depressive symptoms at baseline, which can affect self-directedness (SD) and cooperativeness (C) in the Temperament and Character Inventory (TCI). In another of our previous studies (Prediction of major depressive episodes and suicide-related ideation over a 3-year interval among Japanese undergraduates. PLoS ONE 13 (7): e0201047), we adopted a longitudinal design, which made it possible to compare the prevalence of SI and depressive episodes between two timepoints. However, as the theme of this study (PLoS One 13 (7): e0201047) differed from that of the present study and because we did not analyze character configurations, we decided to analyze the relationship between character configuration and depression and SI in the current work.

 Our present study is novel in that we succeeded in eliminating state-effect bias by excluding participants with depressive episodes at baseline. In the population that was asymptomatic at baseline, we examined whether the prevalence of MDE and suicide-related ideation decreased as the character configuration became more mature. We acknowledge that there are six longitudinal studies with similar themes to that of our previous study (PLoS One 13 (7): e0201047). However, to the best of our knowledge, no study has confirmed the relationship between personality development and the onset of depression and SI in young people such as university students. The present study is novel in that it examines the longitudinal relationship between character configuration and the onset of illness in a young population. Although the present study is using a part of the dataset from our previous study (PLoS One 13 (7): e0201047), the analysis method is completely different, a method that could not be implemented in our previous study (PLoS One 13 (7): e0201047). Suicide among young people (from adolescence to people in their 30s, including university students) is the leading cause of death in Japan, making it an even more important issue than in other regions. Moreover, the issue of suicide among young people is becoming increasingly critical, especially against the backdrop of the COVID-19 pandemic. It is therefore meaningful to exclude some data and conduct a re-analysis from a different perspective.

We have revised our manuscript to highlight the differences between this work and previous studies, the need for re-analysis, and the novelty described above.

Page 2, line 27–28, Methods section in the Abstract: “We previously conducted a study using the same data set; this is a re-analysis of the dataset.”

Page 4 lines 83–96, Introduction: “To address this concern and to more accurately verify whether character configuration can predict the novel onset of MDE and SI, subjects in a depressive state at baseline should be excluded from the sample to prevent data contamination. This exclusion allows us to focus on the new onset of depressive state.

Based on the background provided above, we conducted a re-analysis of the dataset from a prior study,[21] with several methodological differences. Our previous study [20] reported that the prevalence of MDE and SI among university students decreased as the character configuration became more mature. However, because that was a cross-sectional study, the depressive symptoms—which can affect SD and C in the TCI—were not considered. In another previous study,[21] we adopted a longitudinal design that enabled us to compare the prevalence of MDE and SI at two timepoints. However, even in this previous report, we did not control for depressive symptoms at baseline. In the present study, we attempted to control for bias related to state-effect by excluding depressive subjects at baseline”

Page 18, lines 297–304, Discussion: “It should be noted that the present study is a re-analysis study that uses part of a dataset from our previous our study [21]; however, the analysis method is completely different and we use a method in this study that could not be applied in our previous study [21]. Suicide is the leading cause of death in Japan among young people (from adolescence to people in their 30s, including university students). This high occurrence makes it an extremely important issue, compared with other regions. Moreover, the issue of suicide among young people is becoming increasingly important, especially during the current COVID-19 pandemic. Therefore, it is meaningful to conduct a re-analysis from a different perspective.”

Comment 2: There are some methodological issues that need attention. First, only 28% of the university students participated in this study. How were they selected, or did they simply represent a volunteer or convenience sample? Was there some systematic means of recruitment of this 28% of the student population? A related issue is that if the sample recruitment was not randomly or systematically selected, there could be important sampling biases. The limitations part of the Discussion needs to include comments about this issue and how the authors think sampling issues might have affected the study results.

Response: Thank you for your insightful comment. We acknowledge that there are sampling issues that might have affected the study results. Only 28% of the university enrollments participated in this study and it is true that their recruitment was not random or systematic, which means that there could possibly be sampling biases. However, the main purpose of administering the PHQ-9 and TCI is to screen for mental health. After explaining the purpose of the research before the tests were conducted, we asked as many students as possible to participate voluntarily. Even though it is desirable to have as many participants as possible, the number of respondents decreased to 28% due to the research design that required multiple questionnaires and the interval between the tests being as long as three years. Continuous efforts should be made in the future to increase the response rate.

Pages 20, 21, lines 364–373, Discussion: “Third, there are sampling issues that may have affected our results. Only 28% of the university enrollments participated in this study. Further, as we did not use random or systematic sampling, there could possibly be sampling biases. However, the main purpose of administering the PHQ-9 and TCI was to screen for mental health. After explaining the structure and purpose of the research before the tests were conducted, we asked as many students as possible to participate voluntarily. Even though it is desirable to have as many participants as possible, the number of respondents decreased to 28%, due to the research design that required multiple questionnaires and the time interval between the tests being as long as three years. Continuous efforts should be made in future to increase the response rate.”

Comment 3: Also, how were the research instruments administered to the students? Were they confidentially administered? For students reporting current suicidal ideation, how did the study manage their clinical risk?

Response: The tests were administered in a mark-sheet written format. The information obtained from the questionnaires was not disclosed and was managed appropriately at the health care center of Hokkaido University. If severe depressive state or SI were detected, the health care center contacted the students telephonically or via email and started intervention, as these students are considered to be at high risk of suicide.

Page 6, line 134–138, 2.2 Measures in the Methods: “The tests were administered in a mark-sheet written format. The information obtained from the questionnaires was not disclosed and was managed appropriately at the healthcare center of Hokkaido University. If severe depressive state or SI were detected, the healthcare center contacted the students telephonically or via email and started intervention, as these students are considered to be at high risk of suicide.”

Comment 4: Another important methodological issue is the use of a self-report symptom scale. The manuscript mentions the excessive burden of instead using a fully diagnostic instrument, especially with such a large sample. Is such a large sample needed, and could studies be designed using smaller sample sizes determined by power estimates that might allow application of diagnostic tools to yield sufficient analysis? It is unfortunate that the Methods and Results sections characterize the PHQ-9 as “diagnostic,” even though it is acknowledged in the limitations section that it is not a diagnostic instrument. Therefore, the use of the term major depressive episode (MDE) throughout the manuscript represents misleading terminology, and a different term for the positive PHQ-9 screening result is needed. Additionally, the item assessing suicidal ideation in the PHQ-9 in this study is overly broad and includes thoughts of self harm that may include contemplation of or engagement in behavior not intended as life-threatening, such as non-lethal self-cutting gestures. This is another limitation that should be acknowledged and discussed in terms of how it might have affected the findings.

Response: The data used in this study were obtained from previous studies and the sample was not obtained solely for this research. Therefore, the sample size was rather large. We did not select the data further because we considered that there would be less risk of arbitrary data selection by the researcher if we used the existing dataset as it is. 

In addition, the term “MDE” should be used with caution, as you pointed out. To address this point, we revised section 2.2.1 Patient Health Questionnaire-9 to emphasize that MDE was based on the PHQ-9 algorithm diagnosis. Since the original reference related to PHQ-9 (Spitzer et al., JAMA. 1999;282:1737-1744), mentions algorithm diagnosis, the analysis was conducted according to that classification. In the literature, the term major depressive syndrome is used; if MDE is misleading, it can be described as major depressive syndrome. As the purpose of the study was to analyze the data obtained in the previous studies, we did not conduct structured interviews with a reduced sample size.

 Next, we address the issue regarding the term “suicide-related ideation.” 

As explained in the manuscript (background), the main reason for adopting the term “suicide-related ideation” is that it is consistent with the ninth item of the PHQ-9. We consider this term to be appropriate for use as an indicator for detecting suicide risk factors. This is because attempts at self-harm are as strong a risk factor for suicide as suicide attempts and it is difficult to distinguish between clear suicidal intent and modest suicidal intent when injuring oneself. We acknowledge that using the broader term “suicide-related ideation”—which includes self-harm—rather than pure suicidal ideation, may have had an impact. However, compared with previous studies such as the work of Ibrahim (2012), the prevalence of suicide-related ideation among college students in the present study was lower at both T1 and T2; this suggests that there was no over detection in our study.

In accordance with your suggestions, we have added the following to the manuscript.

Page 7, lines 150–154. 2.2.1 Patient Health Questionnaire-9 in the Methods. “It should be noted that the term “MDE” should be interpreted with caution, as it is a classification based on the PHQ-9 algorithm screening tool, and not a diagnosis that has been determined through a structured interview. As this study aimed to analyze the relationships in the data obtained from the screening, we did not conduct structured interviews with a reduced sample size.”

Comment 5: Finally, given that the main purpose of this study was to provide early identification of students at risk for major depressive disorder and suicidal ideation that could potentially be addressed by clinical interventions, it would seem that there should be some statistics provided about what proportions of students with depression or suicidal ideation were identified, with indicators such as sensitivity, specificity, and positive and negative predictive values that can confer concrete clinical utility for these purposes.

Response: Thank you for a meaningful comment. We decided to calculate and describe the sensitivity, specificity, and positive predictive value of depressive episodes and suicide-related ideation using data from both groups with high SD and C and groups with low SD and C.

We have added the following sentences below.

Page 17, lines 285–288. 3.4 Character configurations and vulnerability in the Results. “Lastly, the sensitivity, specificity, and positive predictive value for depressive episode when using the data of SC and sc groups were 0.723, 0.543, and 0.079 respectively. Similarly, the sensitivity, specificity, and positive predictive value for SI when using the data of SC and sc groups were 0.746, 0.544, and 0.079 respectively.”

The detailed review of this manuscript is much appreciated, and we have attempted to answer each of the reviewer’s questions fully. We thank the reviewer and editor for the helpful comments, and we believe that we have now produced an improved and more balanced account of our work. We earnestly hope that the revised manuscript will be acceptable for publication in PLoS One.

I look forward to hearing from you. 

Sincerely,

Nobuyuki Mitsui, M.D., Ph.D.

Assistant Professor

Department of Psychiatry

Hokkaido University Graduate School of Medicine

North 15, West 7, Sapporo 060-8638, Japan

Phone: +81-11-716-1161 (Ext. 5973)

Fax: +81-11-706-5081

Email: nmitsui@med.hokudai.ac.jp

---

## [Editor Report · Decision Letter 1]

28 Apr 2021

Character configuration, major depressive episodes, and suicide-related ideation among Japanese undergraduates

PONE-D-21-04425R1

Dear Dr. Mitsui,

We’re pleased to inform you that your manuscript has been judged scientifically suitable for publication and will be formally accepted for publication once it meets all outstanding technical requirements.  Thank you for your thoughtful revision of the original manuscript, which appropriately addressed the initial issues that needed to be clarified. We hope your work helps to improve the screening and management  of the high suicide risk for university students in Japan and elsewhere,

Kind regards,

C. Robert Cloninger, MD, PhD

Academic Editor

PLOS ONE
---

## [Editor Report · Acceptance letter]

3 May 2021

PONE-D-21-04425R1 

Character configuration, major depressive episodes, and suicide-related ideation among Japanese undergraduates 

Dear Dr. Mitsui:

I'm pleased to inform you that your manuscript has been deemed suitable for publication in PLOS ONE. Congratulations! Your manuscript is now with our production department. 

Kind regards, 

on behalf of

Dr. C. Robert Cloninger 

Academic Editor

PLOS ONE